# Study on the Preparation and Effect of Tomato Seedling Disease Biocontrol Compound Seed-Coating Agent

**DOI:** 10.3390/life12060849

**Published:** 2022-06-07

**Authors:** Yao Zhang, Yingying Li, Sibo Liang, Wei Zheng, Xiuling Chen, Jiayin Liu, Aoxue Wang

**Affiliations:** 1College of Life Sciences, Northeast Agricultural University, Harbin 150030, China; zy13263696020@163.com (Y.Z.); l18645066071@163.com (Y.L.); 13100956039@163.com (S.L.); zhengwei1884565765@163.com (W.Z.); 2College of Horticulture and Landscape Architecture, Northeast Agricultural University, Harbin 150030, China; chenx@neau.edu.cn; 3College of Sciences, Northeast Agricultural University, Harbin 150030, China; 13040216@163.com

**Keywords:** tomato damping-off disease, root rot disease, biocontrol bacteria, biological compound seed-coating agent

## Abstract

Tomato damping-off and root rot are the two most common diseases of tomatoes at the seedling stage. At present, biological compound seed-coating agents are gradually replacing chemical agents in preventing and controlling plant diseases and insect pests, regulating plant growth, and ensuring crop yields. In this study, five biocontrol bacteria (*Bacillus amyloliquefaciens* (Ba), *Bacillus subtilis* (Bs wy-1), *Bacillus subtilis* (WXCDD105), *Pseudomonas fluorescens* (WXCDD51), and *Bacillus velezensis* (WZ-37)), with broad antibacterial spectra were mixed with auxiliary factors (inactive components of seed-coating agent) after fermentation to compound a seed-coating agent. In this study, the formula for a compound seed-coating agent was selected through orthogonal experiment. Gaseous silica was used as a thickener, and gum arabic and sodium dodecylbenzene sulfonate were used as a film-forming agent and dispersant, respectively. The mass of fumed silica, gum arabic, sodium dodecylbenzene sulfonate, and pearlescent powder was 1.3 g, 1 g, 0.05 g, and 0.5 g, respectively. Adding gibberellin can improve the ability of seed-coating agents to promote seed germination and plant growth. This showed high efficiency in preventing and controlling seedling diseases and promoting seedling growth. After 6 days of inoculation with *Pythium aphanidermatum*, which caused tomato damping-off disease, the seedling mortality rate was 26.7% lower than that of the sterile water control, and 20% lower than that of carbendazim. After 21 days of inoculation with *Fusarium* sp., which caused tomato root rot disease, the seedling mortality rate was 44.31% lower than that of the control, and 22.36% lower than that of carbendazim. The plant height, stem diameter, root length, fresh weight, and dry weight of tomato seeds treated with biological compound seed-coating agent were significantly higher than that of the control. We tested the shelf life of the biological compound seed-coating agent, and found that the effect of seed germination and radicle growth did not decrease. This research provides information on the production technology and application of biological seed-coating agents in tomato production.

## 1. Introduction

Tomatoes are widely cultivated across the world because of their good taste, rich nutrition, and large yield. In the process of tomato planting, plants are susceptible to a variety of plant diseases. Tomato damping-off is a common disease at the seedling stage in tomato production [1,2,3]. The disease is commonly known as small foot plague, which is a serious soil-borne disease during the seedling stage. When the soil moisture is high, the mycelium grows very fast, spreads quickly, and will occupy dead seedlings, or even destroy the seeds. The disease is widespread, and poses serious threat all over the world, causing huge economic losses in agricultural production. The pathogen causing the disease is *Pythium aphanidermatum*, which has a wide range of hosts, and can cause seedling damping-off, and plant rhizome- and fruit rot. Tomato root rot is another serious disease at the seedling stage of tomatoes. It is caused by fungi and oomycetes such as *Pythium* sp., *Fusarium* sp., *Phytophthora* sp., etc. [4,5]. Root rot caused by *Fusarium* is easy to occur and spread under the conditions of low temperature and high humidity, or large temperature changes, poor ventilation, and high soil viscosity [6,7]. Root rot seriously affects the development of the tomato root system, leading to weakening or even death of the plant, reducing the yield and quality of tomato crops, and ultimately economic loss.

Biological control refers to the use of the relationship between species to achieve green prevention and control of diseases and pests by protecting and utilizing natural enemies, and/or breeding and releasing dominant natural enemies [8]. It has no pollution impact on the environment, produces no residues, and resistance is relatively more difficult to acquire. Microorganisms colonize the plant rhizosphere, which leads to certain growth promotion effects on plants [9]. Biocontrol bacteria are mostly isolated from soil and plants. The antagonistic mechanisms of biocontrol bacteria include competition, antagonism, and inducing plant resistance. Competition refers to the competition between biocontrol bacteria and pathogens in terms of nutrition, spatial sites, and energy. *Bacillus* grows and reproduces rapidly, quickly seizes the living space, and consumes the carbon source, nitrogen source, and amino acids necessary for the pathogen [10]. The antagonistic effect is that the secondary metabolites produced by microorganisms inhibit the growth of pathogens. Mucronella pink can produce a variety of antibiotics, among which the polypeptide compounds have good antiviral activity [11]. Plants have an innate immune system to resist pathogens. Biocontrol bacteria can protect plants and improve their immune ability at the same time. Through the study of the control of corn leaf spot by *Bacillus terrestris*, it was found that the new protein elicitor ATiEn recombinant protein could induce the accumulation of ROS in tobacco leaves, and induce a plant defense response by activating the expression of SA and JA/ET signal pathways and PAL, a key gene of the phenylpropane metabolism pathway [12]. *Pseudomonas* can promote growth by dissolving phosphorus, potassium, and nitrogen fixation, and secretes extracellular enzymes to resist the invasion of pathogens. Different biocontrol strains have different antimicrobial spectrums, and have different effects on plant growth [13,14,15]. Previous research has shown that the biocontrol bacteria that antagonize *Pythium* and *Fusarium* include *Agrobacterium*, *Bacillus*, *Pseudomonas*, etc. [16,17,18,19,20]. As understanding of biocontrol microorganisms continues to increase, the growth-promoting effects of a variety of biocontrol fungi on plants are being discovered. Studies have shown that the hormones (auxin, gibberellin, etc.) produced by biocontrol strains can be used as plant growth regulators. Some beneficial microorganisms can degrade microelements (phosphorus, nitrogen, potassium, etc.) that are difficult to utilize in the soil, or antagonize pathogenic fungi in the soil to form a healthy soil environment, thereby achieving the purpose of promoting plant growth.

Seed coating is a technique in which several substances, such as nutrients or pesticides, can be added to the seeds via adhesive agents, such as hydrogel or soft agar, to enhance their germination and improve seedling performance [21,22]. After sowing, the seed-coating agent is slowly released into the soil to control various diseases and insect pests and promote plant growth. Its biggest advantage is that it can coat the seeds and will not easily fall off [23,24]. Seed coatings are generally divided into two parts: active ingredients and inactive ingredients. The active ingredients directly affect seeds. Different active ingredients lead to different functions of different seed coatings. The active ingredients of seed-coating agents include microbes, pesticides, micro-fertilizers, and plant growth regulators. Pesticides are generally divided into fungicides and pesticides. Commonly used microorganisms include *Bacillus*, *Trichoderma*, actinomycetes, etc. Micro-fertilizer refers to various trace elements, such as copper, manganese, zinc, etc. Plant growth regulators are natural plant hormones or plant-stimulating analogues. Commonly used growth regulators include gibberellins, cytokinins, and growth retardants. The role of the inactive ingredients of the seed-coating agent is to ensure the film formation of the seed-coating agent, control the release of the active ingredients, maintain the properties of the seed-coating agent, and to have no adverse effects on seed germination. It mainly includes film formers, dispersants, thickeners, and dyes, etc. In recent years, the development of crop seed-coating agents has been very rapid, and a variety of seed-coating agents with different functions have been developed. However, current seed coating active ingredients are, relatively, not very effective, and the prevailing chemical composition is still popular in seed-coating agents. Although compared with spraying pesticides, seed-coating agents use less pesticides and low residues, some side effects caused by long-term use of chemical seed-coating agents cannot be ignored. For example, they initiate drug resistance, endanger natural enemies of pests, and alkaline seed-coating agents change soil pH [25,26]. Microorganisms are the first choice for replacing chemical agents nowadays, and even in the future, because of their non-toxic, high-efficiency, and pollution-free advantages. Therefore, using biocontrol bacteria as the active ingredient of seed-coating agents has a very broad prospect in preventing and controlling plant diseases and promoting plant growth.

The biological seed-coating agents developed at present are mostly used in soybean, corn, cotton, and wheat production, and there is little research on the microbial seed-coating agents used on tomato seeds. The present study aim was to use five species of bacteria that provide broad-spectrum resistance (Ba, *Bacillus amyloliquefaciens;* Bs wy-1, *Bacillus subtilis*; WXCDD105, *Bacillus subtilis*; WXCDD51, *Pseudomonas fluorescens*; WZ-37, *Bacillus velezensis*) isolated from the soil, alongside additives and plant growth regulators, to formulate a compound that can prevent tomato damping-off and root rot; i.e., a microbial seed-coating agent that can promote plant growth. At the same time, the effects of microbial seed-coating agents on tomato seed germination, seedling growth, and root development were tested. Pot experiments were conducted to verify the effects of microbial seed-coating agents on damping-off and root rot. This research provides information on seed coating materials and a practical basis for the use of beneficial microorganisms in preventing tomato damping-off and root rot.

## 2. Materials and Methods

### 2.1. Experimental Strains and Plant Materials

Pathogens *Pythium aphanidermatum* (strain PA-4) and *Fusarium* sp. (strain FS-1), and five biocontrol bacteria (*Bacillus amyloliquefaciens* (Ba, ACCC No.10147), *Bacillus subtilis* (Bs wy-1, ACCC No.10655), *Bacillus subtilis* (WXCDD105, CGMCC No.12496), *Pseudomonas fluorescens* (WXCDD51, CGMCC No.12760), and *Bacillus velezensis* (WZ-37, CGMCC No.15766)) were preserved in our laboratory. Two pathogenic fungi were isolated from the diseased parts of tomato plants by tissue separation method and plate dilution coating method. They were named PA-4 and FS-1, respectively, and were stored on the inclined plane of a PDA-containing test tube. From the tomato planting shed, the plants without disease and vigorous growth were selected, and the strains were isolated and purified by multi-point sampling method from about 10 cm of their rhizosphere. Based on the plate confrontation, with tomato gray mold as the pathogen, five biocontrol bacteria were screened and preserved.

The tomato cultivar “Dongnong 713”, provided by the Tomato Research Institute of Northeast Agricultural University, is susceptible to both tomato damping-off and root rot diseases.

### 2.2. Inhibitory Effect of Biocontrol Strains on PA-4 and FS-1

The inhibitory effect of biological control bacteria on the growth of two pathogenic fungi was identified by dual culture method. A 5 mm-diameter sample of pathogenic fungi was inoculated into PDA (Potato Dextrose Agar) medium, three Oxford cups were placed at near the same distance; 100 μL of single and composite biological fermentation broth was added to the Oxford cups, and the control group was only inoculated with pathogenic fungi. The inhibitory effect of the biological control bacteria on the growth of the pathogenic fungi was observed after 7 days.

Five biocontrol bacterial single and composite fermentation broths were inoculated into 75 mL PDB (Potato Dextrose Broth), forming a 1% inoculum, and then inoculated with 5 mm-diameter pathogenic fungi. The control group was inoculated only with pathogenic fungi, and cultured in a shake flask at 200 rpm for 7 days at 28 °C. Thereafter, the hyphae were collected and dried for weighing. Inhibition rate (%) = (dry weight of hyphae in the control group—dry mass of mycelium in the treated group)/dry weight of the mycelium of the control group × 100%. In each group, this was repeated three times.

### 2.3. Germination-Promoting Effect of Five Isolates and Compound Bacteria on Tomato Seeds

Five biological bacterial fermentation broths (10^9^ cfu/mL of the original bacteria solution) were diluted 200 times, and a group of mixed bacterial solutions (H), and a group of sterile water, were set up. The sterilized seeds were soaked in seven treatments, and after 3 h, the tomato seeds were placed on a plate covered with a layer of sterile filter paper; 30 seeds per plate, soaked with 2 mL sterile water. We recorded the seed germination rate at 48 h, and measured the radicle growth at 4 d. Germination rate (%) = (number of germinated seeds/number of tested seeds) × 100%. In each group, this was repeated three times.

### 2.4. Screening of Five Biocontrol Bacterial Composite Proportions

Five biocontrol bacteria (stock solution 10^9^ cfu/mL) were diluted by 200, 400, 500, 1000 gradients, and the measurement of germination rate of tomato seeds was carried out in an orthogonal test for 48 h. Taking the germination rate as an indicator, the test results were analyzed by range, and the level with the largest K value was selected as the test result for each factor. If the range (R) between the levels of each factor was considerably different, the orthogonal test was continued, with the maximum K value of each factor as the center, until the range becomes stable.

### 2.5. Effects of Different Auxiliaries on the Activity of Five Microorganisms

A total of 16 additives were selected based on their physical and chemical properties (Appendix A). Five bacteria were placed in a shaker at 28 °C and 200 rpm for 120 h. Ba, Bs wy-1, WXCDD51, WXCDD105, and WZ-37 were diluted to 2.3 × 10^8^ cfu/mL, 9.8 × 10^9^ cfu/mL, 1.1 × 10^7^ cfu/mL, 3.9 × 10^8^ cfu/mL, and 2.4 × 10^9^ cfu/mL, respectively. Subsequently, 0.2 g of each additive was added to 1 mL of diluted bacterial solution, sealed, and stored at 25 °C for 7 days, after which 9 mL of sterile water was added, and colonies counted. Changes in the number of bacteria were compared. In each group, this was repeated three times.

### 2.6. S-Compound Microbial Seed Coating Formulation

Based on the influence of different additives on different microbial activities, physicochemical properties of the additives, three film-forming agents, dispersing agents, thickeners, and pearl powders were selected. We added 15 mL of sterile water, 1 g of red pearlescent powder, and appropriate HCl to the nine groups (three repetitions per group). The pH was adjusted to 6.5 with 1 mol/L of HCl. Next, the five bacterial solutions were diluted to the required concentration, 1 mL taken from each, and added to the nine samples. The sterilized seeds were coated, treated with sterile water as the control, and placed on a plate with a layer of sterile filter paper, with 30 seeds per plate. The seeds were soaked with 2 mL of sterile water. The 48 h germination rate was selected from three factors: film-forming agent, dispersant, and thickener. The factor with the largest K value determined the additives required for seed-coating agents. In each group, this was repeated three times.

According to the results of first round of orthogonal testing, the film-forming agent, dispersant, thickener, and red pearlescent powder required by the formula were determined. Taking these four additives as factors with different quality levels, the second round of testing was carried out with an L9 (3^4^) orthogonal table to study the proportion of each additive. First, 10 mL of sterile water and 1 mol/L of HCl was added to the nine samples (three per group), and the pH was adjusted to 6.5. The five bacterial solutions were then diluted to the required concentration, 1 ml taken from each solution and added to the nine samples to coat the sterilized seeds; sterile water was used as the control. Subsequently, 30 seeds were placed on a plate, covered with a layer of sterile filter paper, and soaked with 2 mL of sterile water. Taking the 48 h germination rate as the index, the level with the largest K value in each group was selected to determine the proportion of additives in the seed-coating agent. In each group, this was repeated three times.

### 2.7. Optimization of S-Compound Microbial Seed Coating Formulation

First, 0.1 g of GA_3_ was dissolved in 1 mL of 75% ethanol, and 100 μL, 40 μL, 33.3 μL, and 10 μL of GA_3_ was added to the seed-coating agent to coat the sterilized tomato seeds; the control group was a combination of sterile water and the seed-coating agent. The optimal GA_3_ concentration was selected using the 48 h germination rate, and the radicle growth difference was measured after 4 days.

### 2.8. Growth Promotion of Tomato Seedlings by S-Composite Microbial Seed Coating

The tomato seeds are treated with carbendazim, sterile water, and seed coating, and the treated tomato seeds are sown into a seedling tray containing sterilized soil to keep the water content sufficient. After 15 days, the seedlings were separated and transferred to a nutrition bowl. In total, 15 seedlings were treated, and the root length, stem diameter, and plant height were measured with an MNT-300T zinc alloy digital vernier caliper 21 days after transplanting. After the fresh weight was determined, the plants were wrapped in tin foil and dried in an oven at 60 °C for 3 days to obtain the dry weight of the plants. In each group, this was repeated three times.

### 2.9. Control Effect of S-Composite Microbial Seed Coating on Tomato Damping-Off and Root Rot Diseases

Tomato seeds were separately treated with carbendazim, sterile water, and seed coating, and then sown into seedling trays containing sterilized soil to maintain sufficient water content. After seed germination, PA-4 and FS-1 were mixed at a ratio of 1:10 (*v/w*), and the seedlings were transplanted into the soil of PA-4 and FS-1, respectively. Seedling mortality in PA-4 soil was studied after 48 h. On the 20th day, the incidence of plants in the soil with FS-1 was investigated. According to the root rot disease root stalk area classification: 0, no root rot and brown; 1, root rot accounted for less than 1/3 of the root; 3, root rot accounted for 1/3~1/2 of the root; 5, root rot accounted for 1/2~3/4 of the root; 7, root rot accounted for more than 3/4 of the root. In each group, this was repeated three times.
$$\mathrm{Disease}\;\mathrm{index}\;\left(\%\right)=\frac{\sum \;\mathrm{number}\;\mathrm{of}\;\mathrm{diseased}\;\mathrm{leaves}\;\mathrm{at}\;\mathrm{all}\;\mathrm{levels}\;\times \;\mathrm{relative}\;\mathrm{series}\;\times 100\%}{\mathrm{total}\;\mathrm{number}\;\mathrm{of}\;\mathrm{leaves}\;\mathrm{investigated}\;\times \;9}$$
$$\mathrm{Control}\;\mathrm{effect}\;\left(\%\right)=\frac{(\mathrm{blank}\;\mathrm{control}\;\mathrm{disease}\;\mathrm{index}\text{-}\mathrm{treatment}\;\mathrm{groupdisease}\;\mathrm{index})\;\times \;100\%}{\mathrm{blank}\;\mathrm{control}\;\mathrm{disease}\;\mathrm{index}}$$

## 3. Results

### 3.1. Inhibitory Effect of Five Biocontrol Bacteria on sPA-4 and FS-1

We conducted plate confrontation tests to detect the inhibitory effects of biocontrol bacteria on pathogens. The results show that single and compound biocontrol bacteria Ba, WZ-37, Bs wy-1, WXCDD105, and WXCDD51 showed different degrees of inhibition on tomato mites and root rot. As shown in Figure 1, the bacterial mixture H showed a strong inhibitory effect on the two pathogenic fungi.

Next, we tested the impact of biocontrol bacteria on the growth of pathogenic organisms. The biocontrol bacteria and pathogens were cultured in shake flasks. After collecting the hyphae, followed by drying, and weighing, the inhibition rate of WZ-37, WXCDD105, WXCDD51, Bs wy-1, Ba, and H on PA-4 hypha reached 92.96%, 91.55%, 100.00%, 91.08%, 93.90%, and 100.00%, and the inhibition rate on FS-1 was 100.00%, 96.67%, 95.54%, 94.71%, 97.21%, and 100.00%, respectively (Table 1, Appendix A). We found that the five biocontrol bacteria did not antagonize the growth of each other. After the strains were mixed and cultivated, their original advantages were not affected, and the antibacterial range of the compounded bacteria was more extensive.

### 3.2. Growth-Promoting Effect of Five Biocontrol Bacteria on Tomato Seeds

We tested the growth-promoting effects of various bacterial liquids on tomato seeds. The growth-promoting effect of the strains on seed germination was in the order; WXCDD105 > Bs wy-1 > WZ–37 > WXCDD51 > Ba. The bacterial mixture (H) was the highest, which was significantly higher than that of the sterile water treatment group (CK) (Figure 2A). Radicle growth was measured after 4 days, and the root length of the H treatment and WXCDD105 was higher than the other treatments (Figure 2B). Two sets of experiments show that bacterial mixtures can be used to better promote plant growth.

### 3.3. Screening of the Ratio of Five Biocontrol Bacteria

From the above test, the combined bacterial liquid had the best effect on inhibiting the growth of RS-1, PA-4 and promoting the growth of tomato seeds. Therefore, we continued to screen the compound ratio of the bacterial solution to obtain better results. According to the orthogonal test table (Appendix A), the highest germination rate of tomato seeds treated with different ratios of composite bacterial liquid was in test 10, reaching 55.6%; the lowest germination rate, in test 5, was 26.7%. The germination rate of sterile water was 43.3%. The strain that had a greater impact on the germination of tomato seeds was the concentration of Ba, and each strain selected the level with the largest K value. The dilution multiples of WXCDD51, Ba, WZ-37, WXCDD105, and Bs wy-1 were 500 times, 300 times, 1000 times, 500 times, and 500 times, respectively.

Based on the composite ratio obtained in the first round of orthogonal experiments as the center, the second round of orthogonal experiments were carried out. The highest germination rate was in test 8, at 57.8%, the lowest germination rate was in test 11, at 37.8%, and the extreme difference tended to be stable (Appendix A). From the K value, the dilution multiples of WXCDD51, Ba, WZ-37, WXCDD105, and Bs wy-1 were 500 times, 400 times, 700 times, 700 times, and 400 times, respectively. The plate colonies of the bacterial solutions under the dilution multiples were subsequently counted. The results show that the concentrations of WXCDD51, Ba, WZ-37, WXCDD105, and Bs wy-1 were 10^4^ cfu/mL, 10^4^ cfu/mL, 10^6^ cfu/mL, 10^6^ cfu/mL, and 10^4^ cfu/mL, respectively.

### 3.4. Selection of S-Compound Microbial Seed-Coating Agent

The effects of various additives on different microbial activities were different (Figure 3). Considering the effects of additives on microbial activity, and the physical and chemical properties of the additives, gaseous silica, diatomaceous earth, kaolin, Senegalese acacia, polyvinyl alcohol, pearlescent pigment, sodium lignin sulfonate, agar, and SDBS were selected.

### 3.5. Development of S-Compound Microbial Seed Coating Formulation

We used the above additives to carry out a round of orthogonal tests, and mixed the additives according to Appendix A, i.e., thickener–film former–dispersant–pearl powder in the ratio of 1 g:1 g:0.5 g:1 g. After 96 h, the seed germination rate of test 5 was the highest, at 94.7%, and the lowest was in test 9, which was only 2.7%. It can be seen from the extremely poor R that the key factor affecting seed germination is the thickener. When fumed silica was used as thickener, the relative germination rate of seeds was higher. For the remaining additives, we chose the level with the largest K value in each factor, and the result was that the film-forming agent and dispersant are gum arabic and sodium dodecyl benzene sulfonate, respectively.

As shown in Appendix A, we designed three levels, with the ratio of the first round of orthogonal experiments as the center, to carry out the second round of orthogonal experiments. According to this round of experiments, we selected the highest germination level of each additive. The results obtained are the masses of fumed silica, gum arabic, sodium dodecyl benzene sulfonate, and pearl powder, which were 1 g, 1 g, 0.2 g, and 0.8 g, respectively.

We used the ratio selected by the second round of orthogonal tests as the center to carry out the third round of orthogonal tests to continue to study the seed coating formulation (Appendix A). Based on this round of experiments, the seed coating formulations were preliminarily determined: the masses of fumed silica, gum arabic, sodium dodecylbenzene sulfonate, and pearl powder were 1.3 g, 1 g, 0.05 g, and 0.5 g, respectively.

Tomato seeds were coated according to the ratio of seed-coating agent to seed 2/10 (v/w). After coating, the seeds were pink, and placed at room temperature. It took 20 min for the film to be completely dried.

### 3.6. Optimization of S-Compound Microbial Seed Coating Formulation

Gibberellin is a widespread plant hormone that promotes seed germination and plant growth. Four gradients of GA_3_ were added to the seed-coating agent, the sterile water treatment group (hereafter referred to as CK), and the seed coating treatment group, as the control. After 48 h, the results show that the germination rate of the tomato seed treated with No. 3 was significantly higher than that of others (Figure 4A); the length of the radicle was also longer than other treatments (Figure 4B). These findings shows that adding gibberellin can improve the ability of the seed-coating agent to promote seed germination and plant growth. The seed coating formulation was completed and named S-composite microbial seed coating.

### 3.7. Promoting Effect of S-Complex Microbial Seed Coating on Tomato Seedlings

We tested the growth-promoting effect of compound microbial seed-coating agents on tomato seedlings, using carbendazim (hereafter referred to as D) and CK as the control. The S-treated seeds showed obvious growth advantages after 21 days, and the seedlings developed faster, and the aerial parts were lusher (Figure 5A). Root development was also significantly better than the other two treatments, the main root was longer, and the lateral roots were lusher (Figure 5B). As can be seen from Table 2, the plant height, stem diameter, root length, fresh weight, and dry weight of the S treatment were significantly higher than other treatments. The results indicate that the seed-coating treatment had an obvious promoting effect on tomato seedlings, and plays a role in regulating plant growth.

### 3.8. Preventive Effect of S-Complex Microbial Seed Coating on Tomato Damping-Off and Root Rot Disease

Tomato seedlings were infested with PA-4, and after 96 h, CK seedlings were severely damaged (Figure 6A). The mortality of seedlings in the first 72 h was in the order CK > S > D; after 96 h, the mortality of seedlings was CK > D > S. After 144 h, the seedlings no longer died. At this time, the mortality of CK-, D-, and S-treated seedlings was 90%, 83.33%, and 63.33%, respectively (Figure 6B), indicating that the S-treated tomato seeds had better resistance to tomato damping-off in the seedling stage.

After 21 days of infecting tomato seedlings with FS-1, root rot infestation on the CK group was severe; the roots were all rotted, large disease spots had grown on the main roots, and the lower leaves turned yellow (Figure 6C). The disease index was in the order CK > D > S (Table 3), and the relative disease prevention effect of the D treatment group was 40%, and S reached 84.54%. The above results indicate that the S-compound microbial seed-coating agent had a significantly higher disease prevention effect on tomato root rot than carbendazim.

### 3.9. S-Composite Microbial Seed-Coating Agent Shelf Life Testing

We tested the shelf life of the compound microbial seed-coating agent. The microbial seed-coating agent was stored at 4 °C, and the bacterial content of the seed coating was measured every 30 days. As shown in Figure 7A, the number of bacteria decreased significantly from day 0 to day 30, with a decrease of 23.9%. At this time, the concentration of microorganisms in the seed-coating agent was 5.37 × 10^5^ cfu/mL. On the 90th day, seed-coating agent was used to coat tomato seeds. Compared with sterile water (CK), seed-coating agent (S) still significantly promoted seed germination. The growth of tomato radicles after seed-coating (S) treatment was significantly better than that of CK (Figure 7B,C).

## 4. Discussion

Tomato damping-off disease and root rot are common tomato seedling diseases caused by *Pythium aphanidermatum* and *Fusarium* sp., which can lead to poor plant development and even death [27,28,29]. With the increase in crop multiple cropping index and continuous cropping, the pathogens in the soil continue to accumulate, the types of pests and diseases increase, and this harms the entire growth period of fruits and vegetables. This harm is increasing year by year.

The commonly used control measures include agronomic control practices, chemical control, and physical control. The use of chemical pesticides not only pollutes the environment, but also challenges the survival of organisms and beneficial microorganisms. In addition, it is easy for pests and diseases to become resistant to the chemical compound [30]. Agronomic control and physical control operations are time-consuming and laborious, and the effects are limited. Therefore, to improve soil fertility and crop yield, alternatives to pesticides need to be found; seed coating as a biological control can serve as an environmentally friendly alternative that can effectively improve seed quality, reduce plant diseases, and promote healthy plant growth. Seed coating or pelletizing can also allow beneficial microorganisms to enter the roots, promote nutrient uptake, and enhance tolerance to abiotic stresses [31,32].

*Bacillus* sp. is the most widely used biological agent at present. It is an aerobic or facultative anaerobic bacterium that can produce spores. In agricultural production, extracellular enzymes can be produced to decompose a variety of substances, produce antibacterial substances to resist pathogens, promote plant growth, and induce plant resistance. The preparation of microporous capsules of *Bacillus belesii* NH-1 by immobilization technology not only improved the embedding rate and storage time, but also significantly improved the control effect of *Cucumber Fusarium wilt* [33]. *Bacillus siamensis* was first discovered in 2010 and isolated from *poo-khem* food in Thailand [34]. Researchers obtained a strain of *Bacillus siamensis* CU-XJ-9 from traditional fermented food. Cu-XJ-9 can produce lipopeptide antibiotics, destroy the mycelial structure of *Fusarium graminearum*, and produce nodules, which has a significant antagonistic effect on *Fusarium graminearum* [35]. *Pseudomonas* is a Gram-negative bacterium widely distributed in plant rhizosphere soil and aboveground. This kind of bacteria can reproduce quickly. It is easy to cultivate artificially, in practice, which has the advantages of easy operation, and is an ideal biocontrol factor to inhibit plant diseases. *Pseudomonas* can synthesize and secrete ferritin by itself, increase the microbial availability of soil, reduce the iron concentration in the environment, inhibit the growth and reproduction of pathogenic microorganisms, and achieve the purpose of controlling plant diseases [36].

A seed treatment formulation (flowable concentrate for seed treatment, GIFAP) is a pesticide preparation possessing film-forming properties for crop or other plant seed treatment [37]. For example, *Trichoderma*, a potential biological control agent and growth promoter, can effectively improve plant growth and reduce disease by coating soybean seeds with polymer coatings [38]. Seed coating with molybdenum (B) can promote germination and seedling growth of fine grain aromatic rice cultivar [39]. The growth potential of rice seedlings can be improved by coating seeds with insoluble molybdenum compounds in sulfur-rich submerged soil [40]. Seed-coating agents can be classified into physical type, chemical type, biological type, and specific type according to their use. The microbial seed-coating agent is a seed-coating agent mainly composed of microorganisms and microbial metabolites [41]. This kind of seed-coating agent can effectively prevent crop soil-borne diseases and promote plant growth, does not affect the environment, and is safe for humans and animals [42]. This study combines five biocontrol microorganisms that can control tomato root rot and damping-off disease. We set out to use the broad-spectrum antagonism of five biocontrol microorganisms against tomato root rot and tomato damping-off disease, and report a high antifungal effect in the five biocontrol bacteria’s broth. Research and development of a new type of environmentally friendly microbial seed coating with broad-spectrum resistance is therefore important. The five microorganisms were compounded, and 15 mL of sterile water, 1.3 g of fumed silica, 1 g of gum arabic, and 0.05 g of dodecylbenzene were added. Adding a certain amount of sodium sulfonate, 0.5 g pearl powder, 1 mol/L of HCl (to adjust the pH to 6.5~7), and then adding 33.3 μL of 0.1 g/mL of GA_3_, the control effect of tomato damping-off was 30%, and it was harmful to tomato root rot. The control effect was 84.54%, indicating that the seed-coating agent has broad application prospects.

The choice of additives is the focus of microbial seed coating formulation development [43]. Researchers conducted a series of studies on the auxiliaries of seed coatings, and concluded that excellent auxiliaries can not only improve the suspension of seed-coating agents, but also increase the efficacy of seed-coating agents [44]. Research in the past found that the film can concentrate the active ingredients in seed coating on the seeds and roots, reducing the diffusion of pesticides into the environment and the amount of contact with the target. The film-forming agent of this study was selected from polyvinylpyrrolidone, gum arabic, polyvinyl alcohol, polyethylene glycol, and gelatin [45,46]. It was found that polyvinylpyrrolidone, gum arabic, and polyvinyl alcohol have relatively little influence on the activity of strains. The seed-coating agent can effectively improve the photosynthetic capacity of crops and enhance the resistance of seedlings to adverse environments. Moreover, the indoor antibacterial rate of the seed-coating agent can reach more than 88%. In this experiment, the effects of additives on seed and microbial activity and the physicochemical properties of the additives were selected. Finally, gum arabic was selected as a film former. At the same time, after a round of orthogonal tests, it was found that thickener was the key factor affecting seed germination. When fumed silica was used as thickener, the seed germination rate was relatively high, which may be related to its bulkiness and porosity, and reduces the resistance during seed germination. This is the coating formulation for this experiment. The seed-coating agent formulated according to this formula has better comprehensive performance than other ratios of seed-coating agents, and has no significant effect on seed germination rate and microbial activity.

Carbendazim is a chemical fungicide with a wide range of fungistasis-inducing target species. It has a strong effect on diseases such as rust rot, leaf spot, head blight, among others. It can be used for foliar spraying and seed treatment. Although the effect of carbendazim is good, people pay more and more attention to the residual problems and environmental pollution caused by carbendazim due to excessive application. In this experiment, an S-compound microbial seed-coating agent was used to coat tomato seeds before sowing, with 50% carbendazim (D) as the positive control, and sterile water (CK) as the negative control. The S-compound microbial seed-coating agent promoted tomato seed germination and plant growth after use, and the effect was significantly higher than for sterile water or carbendazim. The field control effect test also proved that S-composite microbial seed-coating agent has a better effect on tomato damping-off and root rot than that of carbendazim. S-composite microbial seed-coating agents can replace the use of the chemical fungicide carbendazim in field applications [47].

The biological seed coatings currently developed are mostly used in rice, soybean, corn, cotton, and wheat production [48,49,50]. There are few studies on microbial seed coatings for tomato seeds. The S-compound microbial seed-coating agent in this experiment had significant promoting effects on plant height, stem diameter, and root development of tomatoes. Significantly higher than the control treatment, the control effect on tomato damping-off disease was 30%, and the control effect on tomato root rot was 84.54%, indicating that the coating has broad application prospects, providing the foundation for large-scale applications in the field.

## 5. Conclusions

The five biocontrol bacteria promote the germination of tomato seeds and the growth of radicles. The combination of WZ-37, WXCDD105, WXCDD51, BS wy-1, and Ba in the ratio of 100:100:1:1 could increase the germination rate of seeds by 4.5% in 48 h.Five additives with little damage to microbial activity were selected, including gas-phase silica, bentonite, diatomite, gum arabic, and sodium lignosulfonate. The preliminary formula of seed-coating agent was determined by orthogonal tests, and the final additives of seed-coating agent were gas-phase silica, gum arabic, and sodium dodecylbenzene sulfonate.Upon addition of 33.3 μL GA_3_ to S-compound microbial seed-coating agent, the effect of improving seed germination rate was the most significant.S-compound microbial seed-coating agent had significant growth-promoting effects on tomato seeds and seedlings. Pot experiments proved that S-compound microbial seed coating had good preventive effect on tomato damping-off disease and root rot, which was better than commercial chemical pesticide carbendazim wettable powder.

## Figures and Tables

**Figure 1 life-12-00849-f001:**
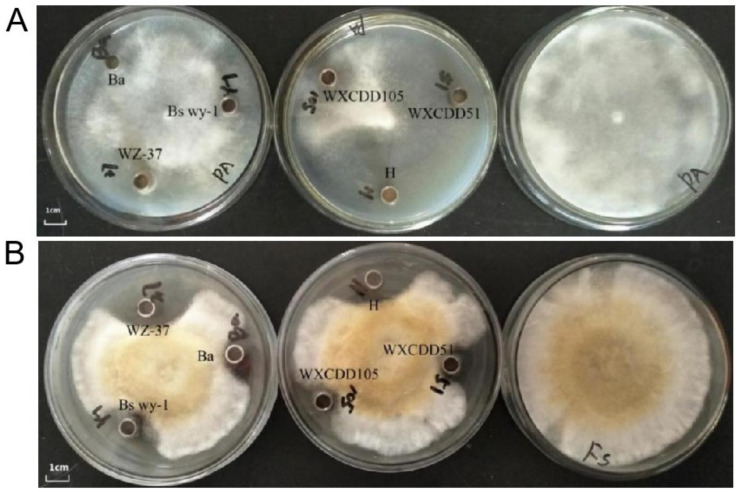
Antimicrobial effect of biocontrol microorganisms on the causative agent of tomato damping-off (**A**) and root rot disease (**B**).

**Figure 2 life-12-00849-f002:**
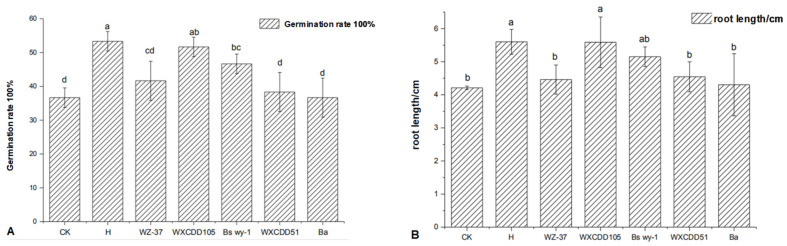
Effects of different bacterial treatments on tomato seed germination (**A**) and the growth of seed radicles (**B**). Note: Different letters indicate significant differences within each category according to one-way analysis of variance (ANOVA) followed by Duncan’s test at the 0.05 alpha level of confidence. Error bars represent the standard deviation of three independent biological replicates.

**Figure 3 life-12-00849-f003:**
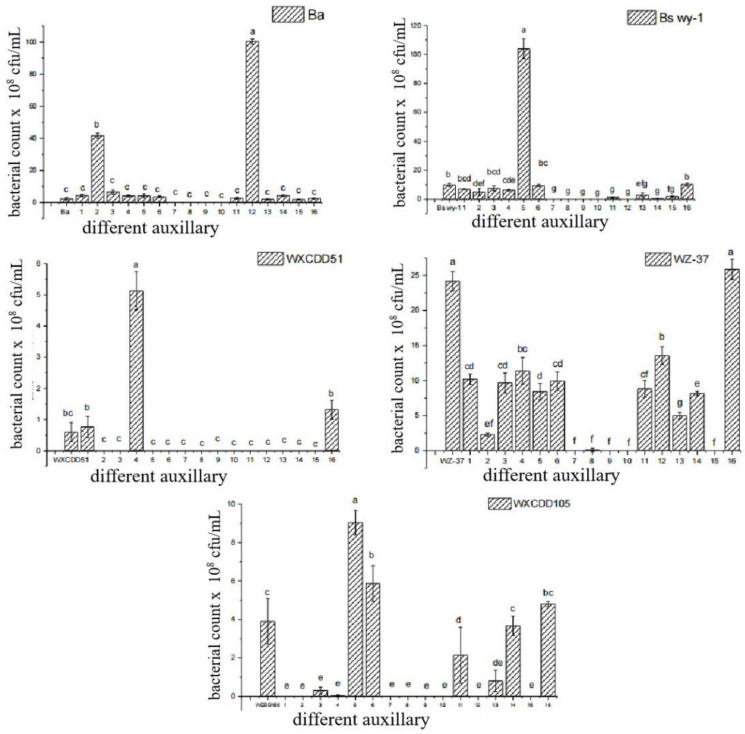
Effect of different auxiliary on microbial activity. Note: Different letters indicate significant differences within each category according to one-way analysis of variance (ANOVA) followed by Duncan’s test at the 0.05 alpha level of confidence. Error bars represent the standard deviation of three independent biological replicates. Auxiliary number: thickener (1. bentonite, 2. dextrin, 3. diatomite, 4. fumed silica, 5. kaolin); dispersant (6. sodium lignosulfonate, 7. SDBS, 8. pull powder, 9. sodium alkyl naphthalene sulfonate, 10. sodium polyacrylate); film former (11. polyvinylpyrrolidone, 12. gum arabic, 13. polyvinyl alcohol, 14. polyethylene glycol, 15. gelatin); indicator (16. pearl powder).

**Figure 4 life-12-00849-f004:**
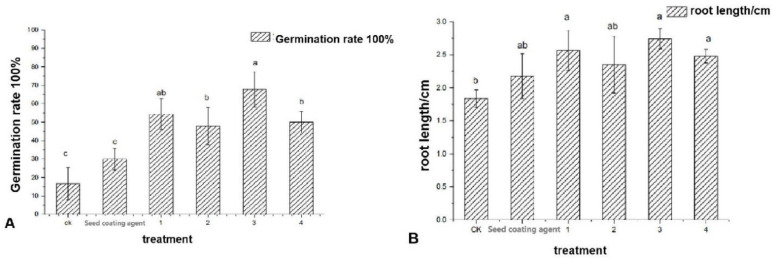
Effects of different concentrations of GA_3_ on tomato seed germination (**A**) and the growth of seed radicles (**B**). Note: CK seed coating represents sterile water, seed-coating agent 1–4 means adding 60 μL, 40 μL, 33.3 μL, 10 μL (four gradients) of GA_3_ to the seed-coating agent. Error bars represent the standard deviation of three independent biological replicates. Different letters indicate significant differences within each category according to one-way analysis of variance (ANOVA) followed by Duncan’s test at the 0.05 alpha-level of confidence.

**Figure 5 life-12-00849-f005:**
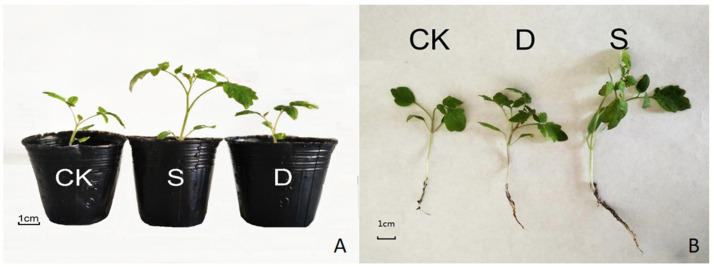
Effects of different treatments on the growth of tomato seedlings: development of above ground parts (**A**); effect on root development (**B**). Note: “CK” means sterile water; “D” means carbendazim; “S” means seed-coating agent.

**Figure 6 life-12-00849-f006:**
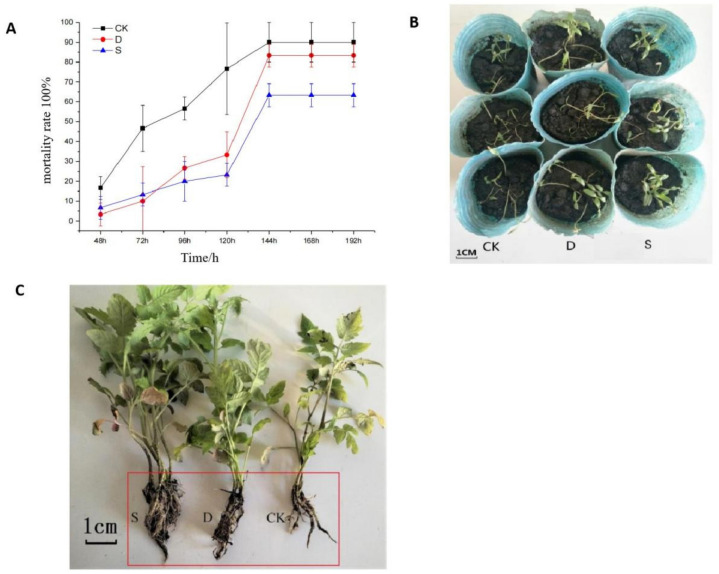
The tomato damping-off disease incidence (**A**) and occurrence tendency (**B**) under different treatments. The incidence of tomato root rot disease under different treatments (**C**). Note: “CK” means sterile water; “D” means carbendazim; “S” means seed-coating agent.

**Figure 7 life-12-00849-f007:**
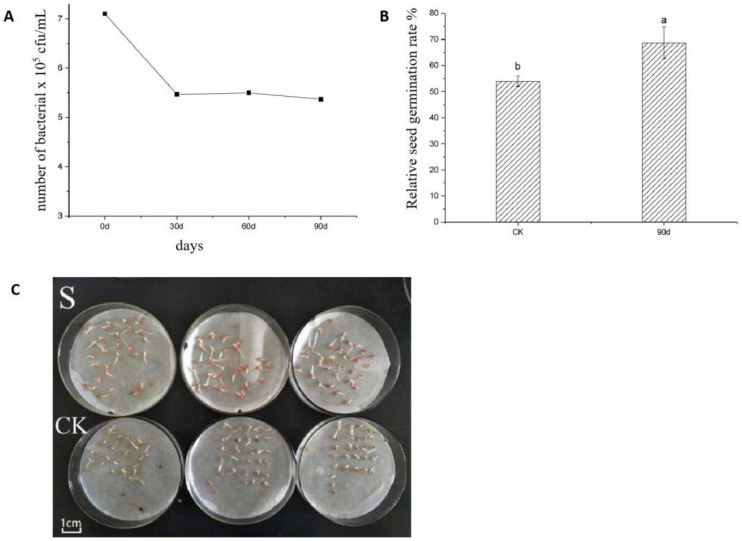
Time course of the microbial content in seed coating formula (**A**), germination rate of tomato seeds (**B**), the growth-promoting effects of S-complex microbial seed coating formula on tomato seed germination after 90 days of storage (**C**). Note: “CK” means sterile water; “S” means seed-coating agent. Different letters indicate significant differences within each category according to one-way analysis of variance (ANOVA) followed by Duncan’s test at the 0.05 alpha-level of confidence.

**Table 1 life-12-00849-t001:** Inhibition of biocontrol bacteria to plant pathogenic fungi.

Biocontrol Strains		PA-4 (*Pythium aphanidermatum)*	FS-1 (*Fusarium* sp.)
WZ-37 (*Bacillus velezensis)*	Dry weight of mycelium (g)	0.015 ± 0.0002c	0.000 ± 0.0000f
	Inhibition rate (g)	92.96%	100.00%
WXCDD105 (*Bacillus subtilis)*	Dry weight of mycelium (g)	0.018 ± 0.0005b	0.012 ± 0.0005d
	Inhibition rate (g)	91.55%	96.67%
WXCDD51 (*Pseudomonas fluorescens)*	Dry weight of mycelium (g)	0.000 ± 0.0000f	0.016 ± 0.0007c
	Inhibition rate (g)	100.00%	95.54%
Bs wy-1 (*Bacillus subtilis)*	Dry weight of mycelium (g)	0.010 ± 0.0003e	0.019 ± 0.0002b
	Inhibition rate (g)	91.08%	94.71%
Ba (*Bacillus amyloliquefaciens)*	Dry weight of mycelium (g)	0.013 ± 0.0002d	0.01 ± 0.0005e
	Inhibition rate (g)	93.90%	97.21%
H (bacterial mixture)	Dry weight of mycelium (g)	0.000 ± 0.0000f	0.000 ± 0.0000f
	Inhibition rate (g)	100.00%	100.00%
CK (sterile water)	Dry weight of mycelium (g)	0.213 ± 0.0008a	0.359 ± 0.0004a
	Inhibition rate (g)	-	-

Note: The result is the average of three biological repetitions. Different letters indicate significant differences within each category according to one-way analysis of variance (ANOVA) followed by Duncan’s test at the 0.05 alpha level of confidence.

**Table 2 life-12-00849-t002:** The promoting effect of seed coating on tomato seedlings.

Treatment	Plant Height/cm	Stem Diameter/mm	Root Length/cm	Fresh Weight/g	Dry Weight/g
CK	7.3 ± 0.70b	1.42 ± 0.03b	4.33 ± 0.23b	1.20 ± 0.12c	0.081 ± 0.01b
D	7.43 ± 0.35b	1.41 ± 0.13b	5.10 ± 0.53ab	1.51 ± 0.06b	0.0903 ± 0.01b
S	9.95 ± 0.73a	1.88 ± 0.15a	6.01 ± 0.28a	2.71 ± 0.08a	0.1727 ± 0.01a

Note: “CK” means sterile water; “D” means carbendazim; “S” means seed-coating agent. Different letters indicate significant differences within each category according to one-way analysis of variance (ANOVA) followed by Duncan’s test at the 0.05 alpha-level of confidence.

**Table 3 life-12-00849-t003:** Control effect of tomato root rot disease under different treatments.

Treatment	Disease Index/100%	Relative Control Effect/100%
CK	52.38 ± 6.44a	
S	8.07 ± 3.59c	84.54
D	31.43 ± 7.42b	40

Note: “CK” means sterile water; “D” means carbendazim; “S” means seed-coating agent. Different letters indicate significant differences within each category according to one-way analysis of variance (ANOVA) followed by Duncan’s test at the 0.05 alpha-level of confidence.

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
