# Peer review of "Study on the Preparation and Effect of Tomato Seedling Disease Biocontrol Compound Seed-Coating Agent"

_life, 2022, doi:10.3390/life12060849_

Round 1
Reviewer 1 Report
The study has some interesting and novel aspects which would be of interest to the international scientific community. However, this manuscript needs significant improvements, both in terms of language and scientific content. Abstract does not do enough to explain which bacteria are used in this study, nor composition/formation of coating agent. Introduction section should provide more background information about antagonistic mechanisms of biocontrol bacteria for improved plant growth and protection, and better explain how they help the plants to establish the resistance against various pathogens. Throughout the text, the authors write about bacteriostatic action, antibacterial action, pathogenic bacteria, although they studied the pathogenic fungi – Pythium and Fusarium. In Material and Methods, it is noticed insufficient information about Bacillus and Pseudomonas strains regarding its origin, methodology of isolation and pathogenicity. The same information is missing for pathogens. On what basis the strains were selected for the study? Have any characteristics of bacterial strains already reported in other publications and are selected Bacillus and Pseudomonas strains commercially used? Authors should use uniform term for coating agent, and better explain how this agent is formed. Subtitles in Materials and Methods, as well as titles of tables and figures, should be rewritten. Simpler terms for bacterial treatments instead of long and confusing group of words (for example, five biocontrol bacteria and complex bacteria; five strains of biocontrol bacteria and compound bacteria, etc.) should be used. Also, authors should write the full scientific names for the used bacteria and fungi followed by the strain code. Authors do not provide sufficient information about replicates derived from different tests. Figures and tables in results do not appear to have statistical information/explanation. Relevance of the extensive study is not properly discussed. Discussion needs to cite other research in this area but the paper and to connect this research to other research in a more meaningful way. It should be focused and improved. Proper conclusions based on the new findings made in the study are missing. Check that all references are cited correctly. I recommend this research for publication after major revision. Accept, but suggest changes to the article as specified in this review. Other comments are included in File.

Reviewer 2 Report
Dear Authors
Please revise you article according to my comments in the attached pdf file of the manuscript.

Reviewer 3 Report
Dear colleagues.
What statistical program was used to process the data (lines 98-146)?
The Supplementary Materials (lines 413-418) speak about the orthogonal text. In my opinion, it is better to specify such details after the description of the methodology.
It is also necessary to specify the number of repetitions of each variant.
The caption to Table 1 (lines 206-207) does not indicate the number of biological repetitions of experiments for which the average was considered.
The same applies to Figure 2-4, Figures 2,3 and some others are too shifted to the left.
It is better to give a transcript of the signatures CK, D, S under Figure 6 (lines 297-300), it is difficult to search for them in the text.
Figures 5 and 7 are again strongly shifted.
Captions to many drawings are illegible, fonts should be enlarged.
Round 2
Reviewer 1 Report
Check again the names of the genera in the manuscript.